# Cannabis and Inflammation in HIV: A Review of Human and Animal Studies

**DOI:** 10.3390/v13081521

**Published:** 2021-08-02

**Authors:** Ronald J. Ellis, Natalie Wilson, Scott Peterson

**Affiliations:** 1Departments of Neurosciences and Psychiatry, University of California, San Diego, UCSD HNRC, Mail Code 8231 220 Dickinson Street, Suite B, San Diego, CA 92103, USA; 2Department of Community Health Systems, School of Nursing, University of California, San Francisco, 1700 Owens Street, Suite 316, San Francisco, CA 94158, USA; Natalie.Wilson@ucsf.edu; 3Sanford Burnham Prebys Medical Discovery Institute, 10901 N Torrey Pines Road, La Jolla, CA 92037, USA; speterson@prescientmetabiomics.com

**Keywords:** cannabis, inflammation, HIV, endocannabinoid system, gut microbiota, gut barrier integrity

## Abstract

Persistent inflammation occurs in people with HIV (PWH) and has many downstream adverse effects including myocardial infarction, neurocognitive impairment and death. Because the proportion of people with HIV who use cannabis is high and cannabis may be anti-inflammatory, it is important to characterize the impact of cannabis use on inflammation specifically in PWH. We performed a selective, non-exhaustive review of the literature on the effects of cannabis on inflammation in PWH. Research in this area suggests that cannabinoids are anti-inflammatory in the setting of HIV. Anti-inflammatory actions are mediated in many cases through effects on the endocannabinoid system (ECS) in the gut, and through stabilization of gut–blood barrier integrity. Cannabidiol may be particularly important as an anti-inflammatory cannabinoid. Cannabis may provide a beneficial intervention to reduce morbidity related to inflammation in PWH.

## 1. Introduction

This brief review will cover potential benefits of cannabis in reducing persistent inflammation and immune activation in virally suppressed people with HIV (PWH) and the possible resulting clinical benefits. While no randomized clinical trials have been performed, both pre-clinical and clinical evidence supports these potential benefits. We discuss sources of inflammation in HIV, their clinical impact, the endocannabinoid system (ECS), effects of exogenous cannabinoids on the ECS and inflammation, particularly neuroinflammation, and potential treatment implications. This review attempts to weave together research threads from multiple areas: clinical, pre-clinical, in vivo and in vitro. The goal is to be integrative, not exhaustive. Overall, the observations reviewed here suggest a program of future basic and clinical research to explore the potential benefits of cannabinoids for the treatment of inflammation-related disorders in PWH. Additional work in this area is reviewed by Yadav-Samudrala and Fitting [1].

## 2. Cannabis and Its Use in People with HIV (PWH)

The proportion of PWH who use cannabis is 2–3 times higher than in the general population (PWH 14–56%, versus people without HIV (PWoH) < 10%) [2,3,4,5,6,7,8,9], in part because many PWH use cannabis to manage symptoms such as nausea, sleep disorders, musculoskeletal and neuropathic pain, anxiety, and depression [10,11]. There is, however, great heterogeneity in findings on therapeutic benefits of cannabis, likely due in part to extensive variation in the formulations being used. Components of cannabis, present in varying degrees depending on the formulation, include the principal psychoactive component, tetrahydrocannabinol (THC), identified in the 1960s, cannabidiol (CBD), cannabinol (CBN), cannabigerol (CBG), cannabidivarin (CBDV), and other compounds, such as terpenes. Each of these exerts unique pharmacological actions, with considerable evidence suggesting that the whole is greater than the sum of its parts (the “entourage” effect), reflecting therapeutic synergies between all of the phytocannabinoids and phytoterpenoids [12,13,14]. This phenomenon may explain why isolated compounds derived from cannabis appear only capable of exerting a limited effect as seen in drugs such as with the synthetic ∆-9 THC drug, dronabinol. To what extent each of these components confers anti-inflammatory benefits is a current focus of research. To what extent each of these components confers anti-inflammatory benefits is a current focus of research.

## 3. Chronic Inflammation in HIV Infection: Presence, Mechanisms and Adverse Impact

An extensive literature beyond the scope of this review demonstrates persistently elevated levels of inflammatory and immune activation biomarkers even in PWH on antiretroviral therapy (ART) who have achieved plasma HIV RNA levels less than 50 copies/mL, designated “virally suppressed” [15,16,17,18]. These include interleukin-6 (IL-6), C-reactive protein (CRP) and tumor necrosis factor alpha (TNFα) [19,20,21,22,23], CD4+ T cell depletion in gut lymphoid tissue persists [24], activated T cells are increased [25], and many immune cells are senescent [26]. PWH also have a dysfunctional gut epithelial barrier (i.e., “leaky gut”), an increase in the permeability of the intestinal barrier accompanied by gut dysbiosis and immune dysregulation from loss of T cells in the GALT, resulting in activation of pro-inflammatory soluble CD14 (sCD14), a response to circulating bacterial products, or microbial translocation [15,17]. The central nervous system (CNS) is an important site of inflammation in HIV, as demonstrated by measuring inflammatory biomarkers in cerebrospinal fluid (CSF) [27,28,29,30], by brain magnetic resonance spectroscopy (MRS) [31], and by positron emission tomography using microglial activation markers [32,33].

## 4. Pathophysiology of Persistent Inflammation in HIV

Several mechanisms contribute to the persistence of inflammation in virally suppressed PWH. These include coinfections and gut dysbiosis. Cytomegalovirus (CMV) co-infection is highly prevalent in PWH; CMV replication is frequently reactivated, triggering an inflammatory response [34]. As mentioned above, gut microbial dysbiosis, which promotes dysfunction of the gut epithelial barrier, resulting in a positive feedback loop sustained by increased microbial translocation of pro-inflammatory antigens such as lipopolysaccharide (LPS) and subsequent immune activation and chronic inflammation [15,17]. Whereas normal commensal flora contribute to tolerance and balance between T helper subsets [35], loss or replacement of these beneficial flora leads to loss of T-helper type 1 (TH1) function, amplifying GALT dysfunction in HIV infection [36]. Depletion of Th17 cells in the GALT leads to reduced IL-22 production, diminishing epithelium repair processes and maintenance of tight gap junctions. Such barrier defects create a pathway for microbial products to escape the gut lumen and enter the systemic circulation. The entry of microbial products into the blood, known as microbial antigen translocation (MAT) triggers the innate immune response and release of pro-inflammatory cytokines, such as interleukin (IL)-1β, TNF-α and others [15,37]. Abundant data show activation of the NLRP3 inflammasome in virally suppressed PWH [38,39].

Hepatitis C virus co-infection is common in PWH and may be another source of microbial translocation that drives inflammation [35,40]. Plasma levels indicative of microbial translocation in HIV-HCV co-infection were higher than in monoinfected PWH virally suppressed on ART [41]. PWH co-infected with HCV had a marked increase in markers of microbial translocation than uninfected healthy controls, whereas the plasma 16S rDNA was relatively similar, suggesting that it is the immune activation that persists as opposed to the circulating bacterial products [42]. Tudesq and colleagues [43] demonstrated for the first time that the plasma 16S rDNA levels increased with the duration of HIV infection in HIV-HCV co-infection, independent of HCV progression.

## 5. Clinical Impact of Increased Inflammation

Increased inflammation in virally suppressed PWH as described above is associated with adverse health outcomes such as myocardial infarction and even death [19,20,21,22,23,44]. Persistent inflammation also affects the central nervous system (CNS), where microglia and astrocytes are chronically activated (Garden 2002, Tavazzi, Morrison et al. 2014 [45,46], producing neurotoxic cytokines that can contribute to cognitive impairment, depression and other adverse outcomes [47,48,49]. Depression is a particularly impactful adverse effect of inflammation in HIV [50,51,52,53]. We recently assessed the relationship of depressive symptoms in PWH to the plasma biomarkers neopterin, soluble TNF receptor type II (sTNFRII), d-dimer, IL-6, CRP, monocyte chemoattractant protein 1 (MCP-1), soluble (s)CD14 and sCD40L. Adjusting for age, comorbidity status, sex, ethnicity, AIDS status, current and nadir CD4, and virologic suppression on ART, factor analyses reduced the dimensionality of the biomarkers, yielding three factors, one of which was loaded on d-dimer, IL-6 and CRP and was correlated with worse depressed mood. We also reported that poorer social support was associated with higher levels of plasma MCP-1, IL-8 and VEGF, as well as CSF MCP-1 and IL-6 (Ellis, Iudicello et al. 2020 [28]), suggesting that that enhancing social support might be an intervention to reduce inflammation and its associated adverse outcomes among PWH.

## 6. The Endocannabinoid System (ECS): Mediator of Cannabis Effects on Inflammation

The ECS comprises a network of receptors, endogenous ligands and enzymes expressed in diverse cell types. Among the many functions of the ECS is regulation of energy use and substrate metabolism to maintain homeostasis [54,55,56]. Components of exogenously administered cannabis bind to EC receptors, thereby modulating the function of the ECS. ECS signaling pathways have been pursued as a target for future pharmacotherapy to reduce inflammation and provide therapy in pathological conditions [55,57].

The cannabinoid receptors type-1 (CB1R) and -2 (CB2R) are expressed in most tissues. CB2Rs are densely expressed in immune tissue and organs in diverse cell types including macrophages, splenocytes, microglia, monocytes, and T-cells resident in the thymus, spleen, and bone marrow and tonsils [58,59,60], providing a mechanism by which cannabinoids can exert anti-inflammatory effects. CB1Rs are most abundant in the brain, where they serve to modulate neurotransmitter activities, thereby mediating effects of phytocannabinoids on neurobehavior [55,56,58]. They are particularly highly expressed in nociceptive areas of the CNS, as well as in the cerebellum, hippocampus, limbic system, and basal ganglia. They are not found in the medullary respiratory centers and thus, unlike opioids, do not cause respiratory depression. CB1Rs are also expressed on immune, cardiac, and testicular cells [61]. In the GI tract, CB1Rs are involved in feeding, gastrointestinal motility, satiety signaling and energy balance [58]. CB1R peripheral activity includes lipogenesis and inhibition of adiponectin, found at elevated levels in obese and diabetic individuals (Nigro, Scudiero et al. 2014, Achari and Jain 2017 [62,63]). CB1R signaling has been linked to increased levels of free fatty acids, low HDL, high triglycerides and insulin resistance [64,65].

The two main endocannabinoids are arachidonoyl ethanolamide, or anadamide (AEA), and 2-arachidonoylglycerol (2-AG), both derived from lipid precursors and synthesized on demand [55,58]. Endocannabinoids in the postsynaptic neuron are released into the synaptic cleft, and travel retrograde to the presynaptic neuron, where they inhibit neurotransmitter release. The principal enzymes for degradation of ECs are fatty acid amide hydrolase (FAAH) and monoacylglycerol lipase (MAGL). Additional EC-degrading enzymes include cyclooxygenase-2 (COX-2), lipoxygenase (LOX), serine esterases and cytochrome P450.

## 7. Cannabinoids Are Anti-Inflammatory

The anti-inflammatory effects of exogenous cannabinoids are mediated by the ECS (Costiniuk and Jenabian 2019), likely through CB2Rs in the periphery that have immunomodulatory functions [61]. Both preclinical and clinical evidence support these anti-inflammatory effects of exogenous cannabinoids, particularly THC and CBD. This may be particularly important in the context of HIV, which is characterized by persistent inflammation as described above. For example, PWH heavy cannabis users had decreased frequencies of T-cells bearing the activation marker HLA-DR+ CD38+ CD4+ compared to non-cannabis-using individuals [66]. Heavy cannabis users also showed reduced frequencies of antigen-presenting cells that produced pro-inflammatory interleukin-23 and tumor necrosis factor-α. In another study, HIV-infected cannabis users had lower IFN-γ-inducible protein 10 (IP-10) levels in plasma [67,68]. In an experimental study examining the interaction between the ECS and cytokine networks in humans, CB1 and CB2 expression were significantly induced by TNF-α, IL-β, and IL-6. CBD may be a particularly potent anti-inflammatory component of cannabis. CBD reduces pro-inflammatory cytokines, inhibits T cell proliferation and reduces migration and adhesion of immune cells [69]. These effects translate to improved outcomes in disease models. Thus, CBD protected against the deleterious effects of inflammation in a viral model of multiple sclerosis [70]. Cannabidivarin (CBDV), structurally similar to CBD, is a non-psychoactive cannabinoid found in cannabis. Very little work has been conducted on CBDV in PWH. In one study, CBDV was safe but failed to reduce neuropathic pain in patients with HIV [69]. In the laboratory, it has been shown that CBDV decreases fat formation and inflammation in human skin cells [71].

Many anti-inflammatory actions of cannabinoids may be mediated through the gut, particularly through stabilization of the gut barrier. The gut barrier is composed of epithelial cells, tight junctions, and a mucus layer. It controls beneficial nutrient absorption and protects against the deleterious invasion of pathogenic bacteria and toxins from the gut lumen into the blood. The ECS, together with the gut microbiota, regulates epithelial barrier permeability [72,73]. In an animal model of HIV, macaques infected with simian immunodeficiency virus (SIV) showed increased markers of inflammation and immune activation in epithelial crypt cells; these markers were reduced after chronic THC administration [74].

## 8. Effects of Cannabis on Neuroinflammation

Exposure to phytocannabinoids may reduce neural injury by decreasing excitotoxicity and neuroinflammation [75,76]. In a large cohort of PWH, we recently reported that neurocognitive impairment (NCI) was less frequent in cannabis users than non-users, regardless of viral suppression [77]. In comparison, cannabis exposure was not related to NCI among PWoH. Unlike many prior reports, this analysis carefully controlled for any non-cannabis substance use disorders, positive urine toxicology for other illicit drugs and any past methamphetamine use disorder, positive breathalyzer test for alcohol, major depressive disorder and HIV disease characteristics. A possible mechanism of the specificity of the benefits of cannabis only for PWH is the anti-inflammatory effect of cannabis, which may be particularly important for PWH who have persistent inflammation despite good antiretroviral treatment.

In contrast to cannabis’s beneficial actions in PWH, research on PWoH typically reports adverse effects on brain development and neurocognition. Examples include attentional and memory deficits, behavioral problems and structural and functional brain changes [78,79,80,81]. The data are particularly concerning for adolescents [78]. It is possible that PWH are less likely to suffer these adverse consequences than PWoH because of the counterbalancing effects of cannabis in reducing neuroinflammation, as we discussed when considering stroke.

In an animal study, euroinflammation, measured as levels of TNFα, IL-1β, IL-6 and MCP-1, was reduced in the striatum of SIV-infected animals treated with THC [82]. Enteroendocrine signaling and the vagus nerve [83] may provide a mechanism through which the gut microbiota may influence the central nervous system. Additionally, signaling through CB1Rs is influenced by *Akkermansia muciniphila* [84] and administration of this organism to obese and type 2 diabetic mice increased intestinal levels of ECs that control gut inflammation and the gut barrier [85,86]. These relationships between the gut microbiota and the ECS may be therapeutically useful. Thus, in zebrafish treated with a probiotic formulation (VSL#3; Lactobacillus spp., Bifidobacterium spp., and Streptococcus thermophilus) for 30 days, gene expression of FAAH and MAGL, the enzymes responsible for degradation of the endocannabinoids AEA and 2-AG, decreased [87]. Thus, probiotic treatment enhanced endocannabinoid signaling and improved gut integrity.

Gut bacteria control the differentiation and function of immune cells in the intestine, periphery, and brain [88,89,90]. There is increasing evidence that gut microbiota and the immune system are critical factors in the pathogenesis of neurodevelopmental, psychiatric and neurodegenerative disease as microbiota immunomodulation orchestrates communication between the gut and brain [91]. Some cognitive domains are subject to immune-mediated CNS injury from HIV-induced microglial activation and contribute to HIV-related cognitive dysfunction [92]. Furthermore, microglia are exquisitely responsive to the gut microbiome and commensal bacteria support the maintenance of microglia in normal homeostasis conditions. When microbiota is absent, microglia lose the ability to mature, becoming defected in differentiation, and function [88]. In a study with germ-free mice, severely defected microglia led to impaired innate immune responses. Recolonization with a complex microbiota environment resulted in partial restoration of normal microglial features [88]. LPS activates microglial cells leading to neuroinflammation and when chronic, is a likely contributor to CNS pathologies, via a leaky gut–brain barrier.

## 9. The Role of Microbial Antigen Translocation (MAT) in Anti-Inflammatory Effects of Cannabis in HIV

Microbial antigen translocation (MAT) refers to the entry of bacterial, fungal and viral components, such as LPS, and metabolites, such as short-chain fatty acids (SCFAs), cross from the gut lumen into the bloodstream. The endogenous cannabinoid, AEA, contributes to the process by which the gut immune system actively tolerates such microbial antigens [56]. In HIV, MAT is associated with monocyte activation and inflammation. Thus, β-D-glucan (BDG) is a microbially derived antigen that serves as one index of MAT.

The anti-inflammatory effects of cannabinoids may be beneficial with respect to HIV reservoirs, which are the principal barrier to HIV cure. We analyzed HIV DNA in blood as a marker of reservoir size in men who had sex with men and initiated ART within a median of 4 months of estimated date of HIV infection. All achieved suppressed HIV RNA within a median of 5 months [93]. Exclusive use of cannabis, as compared to no substance use or use of other drugs, was associated with a faster decay of HIV DNA during suppressive ART. These results are in line with prior reports of reduced HIV replication and cellular infection rate in the presence of cannabinoids in vitro [94,95]. Thus, the potential anti-inflammatory effects of cannabis could translate to a beneficial impact in reducing HIV persistence. However, there is no consistent evidence that cannabis use affects levels of plasma HIV RNA (viral load) [7,96,97,98].

## 10. Cannabinoids and Inflammation in the Brain in HIV

In addition to their expression in the peripheral immune system, CB2Rs are also expressed in the CNS [99,100,101]. In humans, the bulk of CB2R expression is by microglia and astrocytes, consistent with a role in neuroinflammation. Both in vitro studies and animal models show that CB2R mediated anti-inflammatory activity may account for the neuroprotective action of the ECS by decreasing glial reactivity [102]. Both natural and synthetic cannabinoids are neuroprotective after various types of CNS insults, such as stroke [103]. Preclinical models show activation of CBRs can trigger removal of activated immune cells [75], downregulate pro-inflammatory cytokine and chemokine production [104], and inhibit HIV-associated synapse loss and neural injury [105]. In vitro, THC treatment suppresses several pro-inflammatory factors, including TNF-α, IL-6, and IL-8, and decreases monocyte-derived interleukin IL-1ß production and astrocyte secretion of MCP-1 and IL-6 from a human coculture system [106]. We recently reported that more recent cannabis use was associated with significantly lower IL-16 levels in cerebrospinal fluid and lower soluble tumor necrosis factor (TNF) receptor type-II and IP-10 levels in plasma [107]. An additional benefit of cannabis, likely linked to its anti-inflammatory effects, is stabilization of the blood–brain barrier (BBB), which we demonstrated in a separate report, showing that more frequent use of cannabis was associated with better markers of BBB integrity in PWH [108]. Recently, studies have determined that cannabis is associated with reduced markers of immune activation and inflammation in CSF. In sum, there is substantial evidence that cannabinoids display beneficial anti-inflammatory effects that are relevant to HIV infection.

Anti-inflammatory effects of cannabinoids in the brain may translate to clinical benefits, particularly with respect to neurocognition. HIV causes T-cell and monocyte migration to the brain and subsequent interactions with astrocytes and microglia lead to the secretion of neurotoxic cytokines and chemokines [49,109]. These pro-inflammatory factors are linked to worse neurocognitive performance in PWH [110,111,112,113,114,115], and lowering them might benefit neurocognitive function. We showed that lower levels of the monocyte activation marker, monocyte chemoattractant protein type 1 (MCP-1; CCL2) related to better performance in tests of learning ability, and that lower IP-10 also related to better learning as well as delayed recall and motor skills [77]. These cognitive domains frequently show deficits in virally suppressed PWH [116]. However, not all studies support neuroprotective effects of cannabis. For example, a brain diffusion tensor imaging study suggested axonal loss in the uncinate fasciculus, which is involved in verbal memory and emotion, in cannabis users [117]. This study also showed greater than normal age-dependent fractional anisotropy declines in white matter tracts and globus pallidus of cannabis users, suggesting reduced neuronal integrity in these regions.

Numerous reports have suggested a possible link between cerebrovascular disease and cannabis use. For example, a review of 107 case reports over a total of 55 years described strokes associated with intake of both raw and synthetic cannabis [118]. Affected individuals were most frequently young males with chronic tobacco smoking and unusually high levels of cannabis and alcohol consumption just before their strokes [119]. Ischemic strokes and much more common than hemorrhagic strokes with cannabis use. Proposed underlying mechanisms explaining a possible link between stroke and cannabis use, reactive oxygen species (ROS) generation inducing oxidative stress [120], cerebral artery luminal stenosis, cerebral auto-dysregulation, cardioembolism, reversible cerebral vasoconstriction syndrome (RCVS) [121] and angiopathy. None of these reports focused on HIV infection, where effects of cannabis on chronic inflammation may counterbalance adverse vascular effects.

An additional limitation of the historical literature is the many confounds associated with cannabis consumption that themselves are risk factors for stroke. These include concomitant tobacco smoking and intake of alcohol and synthetic cannabinoids, as well as a variety of other comorbidities. In the case of ROS, similar to tobacco smoke, these could be generated as a byproduct of marijuana combustion rather than a specific effect of cannabinoids. Smoking involves inhalation of products of combustion that may be the source of adverse vascular effects. Thus, no strong link between cannabis and stroke has been yet established [122]. These effects may be eliminated when using cannabis in oral (e.g., Dronabinol) or vaporized form [123]. Ongoing trials may be found at clinicaltrials.gov.

Finally, there is considerable evidence that, rather than being a risk factor for stroke, cannabis may be vasculo- and neuro-protective. Thus, cannabinoids may have significant therapeutic value in stroke, as suggested in a recent systemic review and meta-analysis by England et al. [124] showing that all subclasses of cannabinoids, cannabis-derived, synthetic, specific CB1R, and CB2R agonists significantly reduced infarct volume in transient and permanent ischemia and improved both early and late functional outcome in experimental stroke when given after stroke onset [125]. In large mammals, cerebral vessels perfused with cannabinoids demonstrated relaxation rather than constriction [126].

Cannabis disturbs cognition acutely, but its longer-term effects on brain function in HIV are not well understood. While there is limited evidence of increased cognitive impairment in some cannabis-using PWH [127,128], chronic exposure may also reduce inflammation [67], possibly resulting in improved CNS outcomes among PWH. Still, the degree and pattern of cannabis exposure that may be therapeutic, neutral, or harmful is not understood. We hypothesize that an “optimal” level of cannabis exposure will improve some HIV-related outcomes.

## 11. Mechanisms of Neuroprotection of Cannabis

Studies of human and mouse cannabinoid systems in the context of neuroinflammatory exposures show that CB2Rs are highly upregulated during inflammatory insult and their selective activation reduces vascular inflammation, pathological microglial activation and BBB dysfunction [76], thus indirectly decreasing oxidative stress and subsequent cell death [129], and HIV-associated synapse loss [130]. Taken together, this literature cumulatively suggests there may be some therapeutic potential of compounds that target the cannabinoid system through modulation of neurotoxic and inflammatory processes in HIV disease and other neuroinflammatory diseases [131,132].

To evaluate the effects of cannabis use in PWH and people without HIV (PWoH, we assessed biomarkers of neuroinflammation (sCD14 and CXCL-10 in CSF) and BBB permeability (CSF-to-serum albumin ratio (CSAR)) and soluble urokinase plasminogen activator receptor (uPAR), a receptor for uPA, a matrix-degrading proteolytic enzyme that disrupts the basal lamina around cerebral capillaries [89]. We found a statistically significant interaction between HIV serostatus and frequency of cannabis use (total days over the past month) such that more frequent use was associated with lower concentrations of uPAR in CSF in PWH, but not in PWoH. Within PWH, higher CSF uPAR levels correlated with higher CSAR values and more inflammation (higher CSF concentrations of CXCL-10 and sCD14). These findings suggest that cannabis may have a beneficial impact on HIV-associated BBB injury and neuroinflammation, and since BBB disruption may permit increased entry of toxins with consequent CNS injury, these results support the potential therapeutic role of cannabis among PWH and may have important treatment implications for ART effectiveness and toxicity.

In a recent report [93], we hypothesized that more recent cannabis use would be associated with reduced biomarkers of immune activation and inflammation in CSF. This hypothesis was based on previous research demonstrating that selective stimulation of CB2R suppressed neuroinflammation and microglial activation [118,122,123]. We measured a panel of pro-inflammatory cytokines (interleukin (IL)-16, C-reactive protein (CRP), IL-6, CXCL-10, sCD14 and soluble tumor necrosis factor receptor type II (TNFRII)) in CSF and blood plasma in PWH and PWoH who did or did not use cannabis at various levels from none too heavy. Participants were 35 PWH and 21 PwoH cannabis ever users, 15 never users. We calculated factor scores of biomarkers using exploratory factor analysis (EFA) separately for CSF and plasma. We used multiple linear regression to evaluate the association of factor scores with the effects of cannabis use, HIV status and their interaction. Of the three CSF biomarker factors identified, one that loaded on CRP, IL-16 and sTNFRII was associated with more recent cannabis use in both HIV status groups. In plasma, more recent cannabis use was associated with lower values on a biomarker factor loading on sTNFRII and IP-10. Thus, we found recent cannabis use to be associated with lower levels of inflammatory biomarkers, both in CSF and blood plasma, but in different patterns, consistent with compartmentalization of immune effects. Cannabinoids are highly lipid soluble and sequestered in brain tissue, and thus our findings are consistent with specific anti-neuroinflammatory effects that may benefit PWH with or at risk for neurocognitive impairment. However, other studies of cohorts differing in age and HIV disease characteristics (e.g., 16–29-year-old men who have sex with men; PWH without virologic suppression on ART) have reported conflicting findings.

## 12. Summary: Potential Benefits and Risks of Cannabis in HIV

Cannabis as a clinical intervention in HIV disease would be a significant contribution to the field. When ingested, inhaled, or absorbed, THC and CBD, along with other exogenous cannabis components, are anti-inflammatory and counter oxidative stress. Patients report that cannabis has less harmful effects than other drugs.

However, cannabis used improperly can have adverse side effects. People who use cannabis frequently with heavy doses have a higher risk of developing psychiatric symptoms. Long-term users have a reduction in hippocampal volume, affecting memory and verbal learning. In areas with high cannabis use, hospital emergency rooms report increased prevalence of visits due to nausea/vomiting, cardiovascular, psychiatric complaints. Caution is recommended for adolescents using cannabis due to the effect of cannabis on the developing brain. Adolescents (less than 16 years) who used cannabis regularly developed deficits in executive function involved in planning and decision making, as well as memory loss. Cerebral white matter organization is altered, affecting neural communication, potentially leading to higher impulsivity in adolescents.

Further investigations are needed to refine the effects of dose, timing, and cannabis compound on this relationship, which could inform guidelines for safe cannabis use among populations vulnerable to NCI, cognitive decline, and inflammation. Clinical trials are needed to support recommendations to balance the trade-offs between therapeutic benefits and harm.

## Data Availability

Not applicable.

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
