# Peer review of "Cannabis and Inflammation in HIV: A Review of Human and Animal Studies"

_viruses, 2021, doi:10.3390/v13081521_

Round 1

Reviewer 1 Report

This article clearly presents the potential benefits (and their mechanisms) of cannabis use for HIV-infected (especially virally suppressed) through modulation of inflammation. A focus on neuroinflammation is provided.

Line 38 : as long as you quoted mainly studies reporting only data for HIV+, please provide absolute prevalence figure.

Line 46 : your reference link does not work, and reference format is unappropriate. Please provide a valid reference.

Line 48 : please provide a reference for the entourage effect.

Line 57 : you quoted [20-24]. Please check the relevance of references, it seems that it is in reality [24-28]. This +4 delay seems to be persistent throughout the manuscript, and worsen at ref 42-43.

Please define viral suppression as you focus on virally suppressed people.

Line 57 : to reference your sentence, please use references that report levels of biomarkers in HIV+ vs. HIV- and/or HIV+ suppressed vs. HIV+ not suppressed (instead of studies documenting the relationship between biomarkers and clinical outcomes).

Line 58 : in your sentence (and your paragraph), please make it clearer if you refer to HIV infection or HIV viral suppression ("persists" after what ?).

Line 59 : first occurrence of CNS, please define and check all abbreviations.

Line 63 : Check the font of the sub-title.

Line 65 : please evock HCV co-infection, common and pro-inflammatory as well.

Line 67 : please rephrase the sentence, a verb is missing according to me.

From here it became impossible for me to track the quoted references. Please check and re-order references.

Line 72 : please provide a reference at the end of "HIV infection".

Line 80 : please check the font of the sub-title

Line 85 : please add a reference after "outcomes"

Line 88 : please add the reference of your previous work

Line 109 : "where they serve as and modulate other neurotransmitters" is not clear to me.

Line 111-112 : please add a reference for CB1R localization within the brain.

Line 127 : this generalization seems to broad and oversimple to me

Line 137 : Please qualify your affirmation by evocking inconsistent results such as https://doi.org/10.1038/s41598-021-84352-0 or other works.

Line 139-140 : please add a precision on the type of study (in vitro, preclinical...).

Line 161 : FAAH in capital letters please

Line 188 : please put the reference in an appropriate place in the sentence

Line 189 : first occurence of BBB, please define

Line 192 : please indicate if in HIV+ or not.

Line 198 : please indicate if cannabis use directly correlated with biomarkers of inflammation.

Line 199 and whole paragraph : this paragraph is irrelevant to the sub-title it is placed into. Please consider put it elsewhere.

Line 207 : please evock contradictory results regarding cannabis use and viral load

Line 231 : "may translate" may be more appropriate

Line 264 : did cannabis use directly correlate with inflammation ?

Line 295 : please define NCI

Your last quoted reference is #120 while there are 135 references in your bibliography.

Please insert elements of discussion on harmful effects of cannabis on cerebrovascular complications and cognitive impairments.

Author Response

This article clearly presents the potential benefits (and their mechanisms) of cannabis use for HIV-infected (especially virally suppressed) through modulation of inflammation. A focus on neuroinflammation is provided.

Line 38 : as long as you quoted mainly studies reporting only data for HIV+, please provide absolute prevalence figure.

We have added the requested information (PWH 14 - 56%; versus people without HIV [PWoH] <10%)

Line 46 : your reference link does not work, and reference format is unappropriate. Please provide a valid reference.

Link and reference are now fixed.

Line 48 : please provide a reference for the entourage effect.

Done

Line 57 : you quoted [20-24]. Please check the relevance of references, it seems that it is in reality [24-28]. This +4 delay seems to be persistent throughout the manuscript, and worsen at ref 42-43.

The references have been corrected

Please define viral suppression as you focus on virally suppressed people.

This has been clarified: virally suppressed = plasma HIV RNA less than 50 copies/mL

Line 57 : to reference your sentence, please use references that report levels of biomarkers in HIV+ vs. HIV- and/or HIV+ suppressed vs. HIV+ not suppressed (instead of studies documenting the relationship between biomarkers and clinical outcomes).

This has been addressed.

Line 58 : in your sentence (and your paragraph), please make it clearer if you refer to HIV infection or HIV viral suppression ("persists" after what ?).

This has been clarified.

Line 59 : first occurrence of CNS, please define and check all abbreviations.

The acronym has been defined.

Line 63 : Check the font of the sub-title.

The font is as intended. We are happy to implement changes as desired.

Line 65 : please check HCV co-infection, common and pro-inflammatory as well.

This has been clarified in the revised manuscript.

Line 67 : please rephrase the sentence, a verb is missing according to me.

Appropriate changes have been made.

From here it became impossible for me to track the quoted references. Please check and re-order references.

The references have been carefully checked and links from the text to the bibliography have been used to facilitate verification of the correct reference. The journal’s template inserted a secondary numbering that produced confusion. This has been corrected.

Line 72 : please provide a reference at the end of "HIV infection".

Resolved.

Line 80 : please check the font of the sub-title

The font is as intended. We are happy to implement changes as desired.

Line 85 : please add a reference after "outcomes"

Done.

Line 88 : please add the reference of your previous work

The appropriate reference has been provided.

Line 109 : "where they serve as and modulate other neurotransmitters" is not clear to me.

This has been rephrased to clarify. 

Line 111-112 : please add a reference for CB1R localization within the brain.

The appropriate reference has been provided.

Line 127 : this generalization seems to broad and oversimple to me

We have reconfigured this to be more specific.

Line 137 : Please qualify your affirmation by evocking inconsistent results such as https://doi.org/10.1038/ s41598-021-84352-0 or other works.

We have added two references reporting conflicting findings.

Line 139-140 : please add a precision on the type of study (in vitro, preclinical...).

We have specified this with greater precision.

Line 161 : FAAH in capital letters please

Resolved.

Line 188 : please put the reference in an appropriate place in the sentence

Corrected.

Line 189 : first occurence of BBB, please define

Corrected.

Line 192 : please indicate if in HIV+ or not.

Resolved.

Line 198 : please indicate if cannabis use directly correlated with biomarkers of inflammation.

Addressed

Line 199 and whole paragraph : this paragraph is irrelevant to the sub-title it is placed into. Please consider put it elsewhere.

We have revised the paragraph and relocated this material.

Line 207 : please evock contradictory results regarding cannabis use and viral load

We have added a sentence indicating that there is no consistent evidence that cannabis use affects levels of plasma HIV RNA (viral load), with appropriate references.

Line 231 : "may translate" may be more appropriate

Revised as suggested

Line 264 : did cannabis use directly correlate with inflammation ?

This has been addressed in this resubmission.

Line 295 : please define NCI

We have replaced NCI with neurocognitive impairment

Your last quoted reference is #120 while there are 135 references in your bibliography.

Corrected.

Please insert elements of discussion on harmful effects of cannabis on cerebrovascular complications and cognitive impairments.

We have added material describing the literature on cerebral vascular complications of cannabis use. Additionally, we have reviewed the broader issue of cognitive effects of cannabis, beyond the specific circumstances of HIV infection, which were previously discussed in detail.

Reviewer 2 Report

Ronald J. Ellis et.al review the impact of cannabis use on inflammation specifically in PLWH and concluded that Cannabis may provide a beneficial intervention to reduce morbidity related to inflammation in PWH. This manuscript is based on an important topic and the authors have done a great job in conducting such kind of studies. However, following a description of the major issues identified in this manuscript:

  1. In the abstract, Please mention the nos. of literature reviewed to write this, and accordingly please redraft the abstract.
  2. The introduction part is not well described. It should serve the purpose of leading the reader from a general subject area to a particular field of research. Please consider rewriting the introduction section.
  3. The manuscript organization is confused. It seems like all the content of the manuscript comes under the heading introduction. Are all the contents are under the introduction section? Kindly check and please consider including the heading, subheading, and paragraph numbering.
  4. If there is a method section in the abstract, why there is no dedicated method section in the body of the manuscript. Please provide a dedicated method section in the main body of the manuscript.
  5. Since the authors have mentioned in the abstract that they have conducted a review of literature, authors should properly describe how the literature search was conducted? What was the keyword used to search the literature? How many medical databases (like PubMed, web of science, etc.) were used to search the literature? What were the inclusion and exclusion criteria of the article? How the information in this review article were collected? How much literature was reviewed to write this article etc? Please consider explaining these things in a very systematic and transparent way. There should not be any biased in selecting the literature. If there is any biased in selecting the literature, then completely fails to provide a ‘state of the art’ review.
  6. Lack of figures was the main drawback of this paper. please consider including one for a better explanation
  7. There is no conclusion in the main body of the manuscript. There should also be a dedicated section called “conclusion” where all ideas should be wrapped up and leave the reader with a strong final impression.
  8. Lack of proper discussion and conclusion is the major drawback of this review. Please try to provide an in-depth discussion over the published article/study. It seems like the authors have rush things at the end of the article.

Minor comments

  1. There is numerous typo error in the manuscript. Please consider correcting these issues. This manuscript needs editing.
  2. There are lots of wrong references. please consider correcting
  3. The acknowledgment section is missing.
  4. This manuscript is very carelessly written.
  5. In its current form, the study provides little, if any, contribution to scientific advancement over previously published studies on these topics.

Author Response

In the abstract, Please mention the nos. of literature reviewed to write this, and accordingly please redraft the abstract.

This is not a meta-analysis, nor was it planned as a systematic review. No such systematic search was done. We have clarified this in the introduction.

The introduction part is not well described. It should serve the purpose of leading the reader from a general subject area to a particular field of research. Please consider rewriting the introduction section.

Done.

The manuscript organization is confused. It seems like all the content of the manuscript comes under the heading introduction. Are all the contents are under the introduction section? Kindly check and please consider including the heading, subheading, and paragraph numbering. If there is a method section in the abstract, why there is no dedicated method section in the body of the manuscript. Please provide a dedicated method section in the main body of the manuscript.

The subheading labeling and organization comes from use of the journal’s template, which is designed for a report of primary data. Since this is a review manuscript, not a report of primary data, nor a meta-analysis or systematic review, we have removed the confusing subheadings (methods, results, etc).

Since the authors have mentioned in the abstract that they have conducted a review of literature, authors should properly describe how the literature search was conducted? What was the keyword used to search the literature? How many medical databases (like PubMed, web of science, etc.) were used to search the literature? What were the inclusion and exclusion criteria of the article? How the information in this review article were collected? How much literature was reviewed to write this article etc? Please consider explaining these things in a very systematic and transparent way. There should not be any biased in selecting the literature. If there is any biased in selecting the literature, then completely fails to provide a ‘state of the art’ review.

The approach suggested by this reviewer was not used here. Nothing in the abstract or body text implies that there was an attempt to do a systematic, unbiased, comprehensive literature review. In fact, the current science in this area is not well developed enough to support such an approach. Our review is intended only to suggest lines of future research.

Lack of figures was the main drawback of this paper. please consider including one for a better explanation.

This sort of manuscript does not lend itself to a figure, but we would be happy to entertain a specific recommendation.

There is no conclusion in the main body of the manuscript. There should also be a dedicated section called “conclusion” where all ideas should be wrapped up and leave the reader with a strong final impression. Lack of proper discussion and conclusion is the major drawback of this review. Please try to provide an in-depth discussion over the published article/study. It seems like the authors have rush things at the end of the article.

We appreciate the suggestions and have enhanced the summary section (note there is no “conclusion” section).

There is numerous typo error in the manuscript. Please consider correcting these issues. This manuscript needs editing.

See responses to other reviewers.

There are lots of wrong references. please consider correcting.

See response to reviewer #3.

The acknowledgment section is missing.

We have the added material to the acknowledgment section.

This manuscript is very carelessly written.

We trust the revised version addresses this criticism, which was not shared by the other reviewers.

In its current form, the study provides little, if any, contribution to scientific advancement over previously published studies on these topics.

We are disappointed that this reviewer did not appreciate the value of our review, though we appreciate the positive feedback from the other reviewers. We hope that our enhancements have improved its value to the field.

Reviewer 3 Report

The review by Ellis and colleagues presents a brief summary of the current literature on cannabinoids’ capacity to reduce HIV-related inflammation. The review is well-written, however some aspects of the current literature are notably absent. (1) Dronabinol is one of the most well-characterized compounds clinically and authors are remiss to include it. (2) Less well-characterized, but clinically assessed is cannabidivarin (CBDV). It is unclear why authors omitted CBDV when they focused on other minor cannabinoids.  (3) As well, some of the negative literature (many RCT’s report no improvement in endpoints that may be related to inflammation such as pain) and potential limitations are missing or not well-described. There are many demonstrations of potential cognitive side effects associated with cannabinoid use and this may offset some of the potential benefits. (4) Lastly, it would be useful if authors commented on findings that utilized acute vs. chronic Cb exposure, given that this may account for differences in potential benefits.

Author Response

The review by Ellis and colleagues presents a brief summary of the current literature on cannabinoids’ capacity to reduce HIV-related inflammation. The review is well-written, however some aspects of the current literature are notably absent. (1) Dronabinol is one of the most well-characterized compounds clinically and authors are remiss to include it. 

Response: "Dronabinol is an oral form of pure THC and as such was discussed in the original manuscript, particularly its effects on inflammation as compared to CBD. The manuscript now includes additional details about THC, for example, concerning its contribution to the cannabis “entourage” effect and the potential benefits of oral THC administration relative to smoking cannabis.”

(2) Less well-characterized, but clinically assessed is cannabidivarin (CBDV). It is unclear why authors omitted CBDV when they focused on other minor cannabinoids. 

Response: The following has been added to the manuscript: “Cannabidivarin (CBDV), structurally similar to CBD, is a non-psychoactive cannabinoid found in cannabis. Very little work has been conducted on CBDV in PWH. In one study, CBDV was safe but failed to reduce neuropathic pain in patients with HIV (Eibach, Scheffel et al. 2021). In the laboratory, it has been shown that CBDV decreases fat formation and inflammation in human skin cells (Olah, Markovics et al. 2016).

(3) As well, some of the negative literature (many RCT’s report no improvement in endpoints that may be related to inflammation such as pain) and potential limitations are missing or not well-described. There are many demonstrations of potential cognitive side effects associated with cannabinoid use and this may offset some of the potential benefits.

Response: See response to comments from Reviewer #1 above.

(4) Lastly, it would be useful if authors commented on findings that utilized acute vs. chronic Cb exposure, given that this may account for differences in potential benefits.

Response: The following has been added to the manuscript: “Cannabis disturbs cognition acutely, but its longer-term effects on brain function in HIV are not well understood. While there is limited evidence of increased cognitive impairment in some cannabis-using PWH(Lorkiewicz, Ventura et al. 2018, Okafor, Plankey et al. 2019), chronic exposure may also reduce inflammation(Rizzo, Crawford et al. 2018), possibly resulting in improved CNS outcomes among PWH. Still, the degree and pattern of cannabis exposure that may be therapeutic, neutral, or harmful is not understood. We hypothesize that an “optimal” level of cannabis exposure will improve some HIV-related outcomes.

Round 2

Reviewer 2 Report

The authors have addressed my criticism upto certain extent.

  1. However, in this current form, please take care of the duplicative references, which I have seen lots of duplication for examples line 599-604, 643-646, and lots more. Please format the references according to journal style as well.
  2. The current form of this manuscript still needs some deep editing. please revisit again. Consider correcting the typo errors (as I have already suggested)
  3. Please consider providing if not conclusion then provide a dedicated section like future perspective/future research (as I have already suggested to add a conclusion).
  4. Please summaries the last section “Potential benefits and risks of cannabis in HIV in a table formats.

Reviewer 3 Report

Authors have adequately addressed this reviewer's prior concerns and the manuscript is improved in its coverage of the current state of the field.